# Disaster Chain Analysis of Avalanche and Landslide and the River Blocking Dam of the Yarlung Zangbo River in Milin County of Tibet on 17 and 29 October 2018

**DOI:** 10.3390/ijerph16234707

**Published:** 2019-11-26

**Authors:** Huicong Jia, Fang Chen, Donghua Pan

**Affiliations:** 1Key Laboratory of Digital Earth Science, Institute of Remote Sensing and Digital Earth, Chinese Academy of Sciences, Beijing 100094, China; jiahc@radi.ac.cn; 2University of Chinese Academy of Sciences, Beijing 100049, China; 3Hainan Key Laboratory of Earth Observation, Institute of Remote Sensing and Digital Earth, Chinese Academy of Sciences, Sanya 572029, China; 4Department of Integrated Disaster Risk Reduction, Ministry of Emergency Management of the People’s Republic of China, Beijing 100021, China; eliteast@mail.bnu.edu.cn

**Keywords:** Yarlung Zangbo River, disaster chain, avalanche, landslide, river blocking dam, Qinghai–Tibet Plateau

## Abstract

As a “starting zone” and “amplifier” of global climate change, the Qinghai–Tibet Plateau is very responsive to climate change. The global temperature rise has led directly to an acceleration of glacial melting in the plateau and various glacier avalanche disasters have frequently occurred. The landslide caused by glacier avalanches will damage the surrounding environment, causing secondary disasters and a disaster chain effect. Take the disaster chain of the Yarlung Zangbo River at Milin County in Tibet on 17 and 29 October 2018 as an example; a formation mechanical model was proposed. The evolution mechanism for the chain of events is as follows: glacial melt → loose moraine deposit → migration along the steep erosion groove resulting in glacier clastic deposition then debris flow → formation of the dam plug to block the river → the dammed lake. This sequence of events is of great significance for understanding the developmental trends for future avalanches, landslides, and river blocking dam disasters, and for disaster prevention planning and mitigation in the Qinghai–Tibet Plateau.

## 1. Introduction

In recent years, due to the intensification of natural variability caused by global climate change, the global natural disaster activities have becoming more and more frequent and severe. Along with rapid population growth, the continuous development of the social economy and acceleration of urbanization, strengthened natural disasters have significant impacts on production, daily life, and socioeconomic development [1,2,3]. The chain rule makes this effect much more significant. So the disaster chain has become a hotspot in the disaster research field [4,5,6,7].

The concept of a disaster chain was first developed from the basic theory of disaster science, which was proposed by Chinese seismologist Guo Zengjian in 1987 [8]. The disaster chain was considered to be a series of disasters. Researchers have different understandings on the concept of disaster chains. Feng [9] believes that the disaster chain is a series of disasters caused by primary disasters and one or more secondary disasters within a complex disaster chain transmission relationship, which is affected by various environmental factors. Ni [10] defined the disaster chain as a continuous or synchronous disaster chain sequence, which consists of two or more disasters through causality, homology, and so on. Menoni [11] substituted the concept of chain of losses and failures to the simple couple of hazardous events-damages. In the journal Nature, from the perspective of disaster chain response, Helbing [12] pointed out that “there is usually a causal relationship between disasters, which makes the complexity of the disaster system deepen”. Carpignano [13] stated that the domino phenomenon is caused by the stimulating effect between disaster events that constitute a disaster chain.

In order to prevent the occurrence of disasters and minimize the risk of disasters, it is necessary to understand the evolutionary laws of the occurrence, development, and transformation of disaster chains [14]. Shi et al. [15] realized the definition of disaster as a result of the comprehensive effect of the disaster-causing environment, the hazard factor, and the disaster-affected body from the perspective of geography, and he considered that the disaster chain is a complex disaster system. Xiao [16] and Liu [17] gave a definition of the disaster chain from the perspective of a mathematical level, and they further analyzed the relationship between the internal structural elements of the disaster chain. In terms of the transmission effect of disaster chain, Wen [18] proposed the concept of disaster chain transmission. Follow-up scholars further developed the concept of disaster formation ring, triggering ring, damage ring, and chain-breaking ring in the process of disaster chain transmission, which promoted the theoretical study of the chain characteristics of the disaster chain [19].

At present, the researches on disaster chains, a complex disaster system, are still at the early stage. Generally, the mathematical methods like statistical analysis are used to discuss the disaster chain from different aspects such as risk assessment, disaster loss estimation, and chain-cutting disaster mitigation. For example, through the establishment of a structural graph of a powerful earthquake disaster chain, the Bayesian networks (BNs) are used for the analysis of the key hazard factors affecting life safety in the process of earthquake disaster chain transmission [20]. In the risk assessment of disaster chain, some experts have established models based on complex networks for the risk assessment of typhoon disaster chain [21,22]. Asghar [23] proposed a conceptual model of disaster management. Cyganik [24] and Haddow [25] proposed a disaster chain management procedure from the stages of mitigation, preparedness, response, and recovery. However, the chain reaction processes were not considered. It is impossible to capture the evolution process of disasters to respond to the unknown disaster events in a timely manner. Therefore, the management of disasters would be delayed. May [26] proposed the concept of “cascading threat modeling” to establish a causal relationship model between disaster events. At present, the research on the disaster chain focuses on risk assessment and early warnings of the disaster chain. On the other hand, the existing literature has shortcomings in the semantic association and dynamic description of the evolution process of the disaster chain. Most of them simply simulate the evolution process of disasters from the perspective of visualization [27,28], and they mainly focus on simple spatial and temporal phenomena [29]. Thus research on modeling the evolution process of disaster chains is still lacking.

The high mountain hazard chain is a catastrophic phenomenon of one mountain hazard triggering other mountain hazards under the action of inducing disaster factors. It often consists of two or more high mountain hazards such as debris flow, flash flood, landslide, collapse, ice avalanche, snow avalanche, and soil erosion [30,31,32,33]. The high mountain hazard chain is the result of mass transfer and energy conversion of mountain hazards [34]. Although there are some studies on mountain disaster chains, scholars have also explored some prevention and control methods, such as chain-cutting disaster mitigation from gestation source, and the prevention and control of disasters in the chain [35,36].

Landslides and damming the Yarlung Zangbo River at Milin County in Tibet, are two typical cases that occurred on 17 and 29 October 2018. In the Sedongpu Basin, two landslides blocked the river within 13 days, and the river burst its banks after two days (Figure 1). The dammed lake constitutes a water body with a certain amount of solid matter blocked by mountain river valleys or rivers. In general, a dammed lake refers to a naturally formed water body. These dammed lakes are widely distributed throughout the world, especially in mountain valleys where earthquakes occur frequently [37]. There are many barrier lakes at different scales in China’s mountainous areas, and breaches from some of these dammed lakes have resulted in serious environmental disturbances. China’s barrier lakes are mainly distributed in alpine valleys and alpine regions, and some large earthquakes have resulted in the failure of these barrier lakes [38,39,40,41]. In particular, high mountain and canyon areas with steep and rugged terrain, which experience new tectonic movements, including earthquakes, are active, such that landslides and mudslides have become more frequent [42]. Once large-scale landslides and mudslides occur, blockage of the main river and formation of a barrier lake occurs. Frigid and high-altitude areas tend to be the areas where modern glaciers and glacial lakes develop, and many moraine-dam lakes are known. Between 1949 and 1994, there were 78 incidents of landslides which blocked rivers in mainland China, mainly in the southwestern mountains of Sichuan, Yunnan, and Tibet. There was also a small number of such occurrences in Hubei, Guizhou, Hunan, Shaanxi, Gansu, Qinghai, and Xinjiang [43,44,45]. In 2008, the Wenchuan earthquake gave rise to the formation of 256 barrier lakes, which were mainly distributed along the main rupture zone of the earthquake and 85.6% of the dammed lakes were distributed in the three major fault zones of Longmenshan and within 10 km of the fault [46].

The objective of this study is two-fold: (1) to analyze the “avalanche-landslide-debris flow-barrier lake” hazard mechanism and derivative disasters chain based on disaster system theory, and (2) to summarize the evolution process of the disaster chain and the organizing framework of chain-cutting disaster mitigation. These contributions may provide new insights into the disaster chain prevention and mitigation in the Qinghai–Tibet Plateau and other high mountain environments.

## 2. Analysis of the Disaster Chain Process

### 2.1. Study Area and Data

As a “starting zone” and an “amplifier” of global climate change, the Qinghai–Tibet Plateau is very responsive to climate change [47]. The district is characterized by its complex geology, high in-situ stresses, and earthquakes, repeating ground freeze and thaw, intensive precipitation, frequent landslides, and chained disasters [36,37,38,43,44,45]. The global temperature rise leads directly to an acceleration of glacial melting in the plateau and frequent occurrence of glacier avalanche disasters. Since 1961, the annual average temperature of the Qinghai–Tibet Plateau has increased significantly, with an average increase of 0.36 °C per 10 years [48]. The warming rate at the Tibetan Plateau is twice the rate of global warming. The Tibetan Plateau’s climate and environment have undergone significant changes and the consequences of these changes are evident. The dry and wet seasons of the plateau are distinct, with the rainy season being from May to October (wet season) and the dry season from January to April. The plateau has become “wet”, with an increase in precipitation of 4.5 mm per 10 years in the rainy season and 1.8 mm per 10 years in the dry season [49]. Analysis of remote sensing monitoring data shows that the overall area of the Purog Kangri Glacier has been reduced significantly over the period 1973 to 2016. In 2016, the area of the Purog Kangri Glacier was 389.0 km^2^, a decrease of 17.9% compared with 1973 [50]. The glacier change has been largest in the north, followed by the southeast, with the smallest change being in the west.

The thickness of the active layer in the permafrost regions in the Qinghai–Tibet Plateau has increased with the frozen soil degrading significantly. From 1981 to 2016, the thickness of the active layer increased significantly, with an average thickness of 18.9 cm per 10 years [51]. Since 1961, the maximum depth of frozen soil with an altitude above 4500 m has been the most obvious sign, with an average decrease of 15.1 cm per 10 years. The comparable thickness of the medium-altitude area, an altitude of 3200 to 4500 m, was 4.6 cm per 10 years; and for altitudes below 3200 m the decrease in thickness was even less. In 2016, the maximum depth of frozen soil in the high-altitude areas was the lowest since 1961, which was 91 cm less than the average annual value [52].

The Yarlung Zangbo River is an international river and discharge of the barrier lake will affect downstream regions in India. Inappropriate management of the river may lead to international disputes. Moreover, disaster emergency control measures may be severe, population control, transfer and resettlement are usually challenging, and the rise and fall of the dam water would likely cause secondary disasters, which would seriously threaten the upstream and downstream residential areas and major infrastructure.

The analyzed data were retrieved from the digital elevation model (DEM) data of the study area with 30 m resolution (ASTER GDEM data from the National Aeronautics and Space Administration, USA), the China Province Administrative vector map, the earthquake distribution map (China Earthquake Administration), daily precipitation data download from the China Meteorological Data Sharing Service System (http://cdc.cma.gov.cn/) of Milin County for 1979–2018, and the remote sensing data provided by the Ministry of Natural Resources of China and Ministry of Emergency Management of China.

### 2.2. Disaster Body Formation Process

Early in the morning of 17 October 2018, a landslide occurred in Jiala village, Pai Town, Milin County, Nyingchi City, Tibet. A dam lake was formed in the Yarlung Zangbo River. The water level of the barrier lake rose rapidly at a rate of 0.8 to 1 m/h. Using three-dimensional (3D) remote sensing analysis along the river, the landslide dam was noted to be about 3500 m in length, 415–890 m wide, 77–106 m high, and a volume of the soil and dirt blocking the river was 30 million m^3^ [53,54]. The maximum water depth of the barrier lake was 77 m and the water storage capacity was about 320 million m^3^ [55,56].

The maximum elevation of the debris flow formation zone was about 4100 m, the debris flow circulation zone was about 8.3 km long, and the average gradient ratio was about 21%. The basic composition of the sediment deposit material was gravel and gravel soil, and the top elevation was 2830–2860 m. The bottom elevation was about 2750 m, the flow direction was about 2.4 km long, the width was about 850 m, and the estimated volume was about 40 to 60 million m^3^. The maximum water depth before the dam was about 79.43 m, the return water length was 26.0 km, the natural channel ratio in the backwater area was about 3.05%, and the natural drop in the backwater area was about 79.3 m. The return water level was 2840 m, and the estimated storage capacity was 605 million m^3^. The damming water body began to naturally overflow at 13:30 on 19 October, and the flow rate gradually increased. The estimated maximum instantaneous flow rate was 32,000 m^3^/s. At 21:30 on 19 October, the flood reached the Dexing Hydrological Station in Medog County (168 km downstream of the dam body). At 23:40 on 19 October, the peak water level appeared. The maximum water level increased by 19.76 m, and the corresponding flow rate was 23,400 m^3^/s. By 16:00 on 20 October, the flow rate was 3030 m^3^/s. The water storage capacity of the dammed lake which passed through the Dexing station section was 580 million m^3^, the water level of the dammed lake had dropped by 72 m, and the amount of water entering and leaving the lake had reached equilibrium (Figure 2).

In the early morning of 29 October, the Yarlung Zangbo River dammed lake from 17 October formed a glacial debris flow due to the melting of ice debris and other ice bodies, which caused the Yarlung Zangbo River to stop flowing and form a dammed lake again. Using 3D remote sensing measurement data analysis, the dam body was about 3500 m long, 415–890 m wide, 77–106 m high, and the total volume was about 30 million m^3^. At 9:30 on 31 October, the landslide dam naturally overflowed, and the maximum overflow of the landslide dam was estimated to be about 18,000 m^3^/s at 12:30. At 18:30, the flood passed downstream through Medog County, and then the water level turned back. At 9:00 on 1 November, the water level downstream in Medog County fell and returned to normal. At 18:30, the peak flow passed Medog County, and then the water level started to fall back. At 9:00 on 1 November, the water level in Medog County returned to normal levels.

## 3. Analysis of the Causes of the Disaster Chain of Events

### 3.1. Analysis of the Hazard Factors Affecting the Disaster Chain

The Yarlung Zangbo River barrier lake was caused by landslides and mudslides blocking the river. The dammed lakes occurred in an alpine valley area where the terrain is steep. The height difference in the terrain is higher than 1000 m and the potential energy is very large (Figure 3). It is relatively easy for a high-level landslide to occur in this region. The landslide can slide for several km or even 10 km, forming a dammed lake in the river.

The water storage capacity of the two barrier lakes reached 100 million m^3^ in just several to tens of hours, and the growth rate was rapid, mainly due to the large river flow in the area where the lake was affected. According to the precipitation data of the meteorological department, the recent precipitation in the southeastern part of the Qinghai-Tibet Plateau was significantly higher than that in the same period of the previous year, and in some areas it was four times greater. From May to October 2018, the precipitation at Milin was 422.9 mm, which was 28.7% less than that for a typical year (593.1 mm). Since May, the precipitation in May–August and October was less than 100 mm; among these months, the precipitation in June, July, and August was 40–50% less than normal. However, in September, more precipitation occurred, in excess of 50% of normal levels (Figure 4).

From geological considerations, the barrier lakes are located on the eastern edge of the Qinghai–Tibet Plateau where earthquake activity is strong. On 22 August 2013, a magnitude 6.1 earthquake occurred at the junction of Zuogong County and Mangkang County in Changdu City. On 18 November 2017, a magnitude 6.9 earthquake occurred in Milin County, Nyingchi City. The Milin earthquake triggered the bottom of the Gyala Peri glacier in the northern part of the Yarlung Zangbo River to become mobile and active, and movement sped up [57,58]. Coupled with recent climate change, the melting of glaciers or ice lakes resulted in a huge amount of sediment being deposited in the barrier lake. Given the geological structure, topography, and climate of the region, local natural weathering is a serious factor and landslides often occur.

There have been 22 earthquakes of magnitude 3 or higher since 2012 in the vicinity of the Sedongpu Basin (Figure 5). On 18 November 2017, the epicenter of the Milin 6.9 earthquake was about 10 km away from the mouth of the Sedongpu Basin and the focal depth was 8 km [59]. This earthquake to some extent destroyed the stability of the glacier structure, the accumulations in the valleys, and the moraine.

There are many ice crevasses in the middle of the glaciers and the ends of the glaciers are covered by moraine. The glaciers over the entire valley melted considerably. In 1970, the glaciers of the Sedongpu Basin were an integrated whole. At present, the glaciers are divided into many branches due to strong ablation. The Sedongpu Basin is about 2100 m long and 115 m wide [45]. According to a comparative analysis of remote sensing images for 26, 27, and 30 October 2017 glacial meltwater was the main cause of this debris flow (Figure 6).

### 3.2. Long-Term Accumulation of Rock Debris

As early as 2015, satellite imagery showed that the surface of the slope near the Jinshajiang River landslide point was deformed and broken, and the rock mass slippage was clear. The median value of the average displacement of the landslides was 5.4 m from March to May 2018, 9.3 m from May to July, and 11.8 m from July to September. The top of the Yarlung Zangbo River landslide dam had a large landslide from 5 January 2016 to 8 June 2018 [37,40]. The snow-covered sediment was transported to the Yarlung Zangbo River, partially deposited in the middle of the mountain. Since the lower mountain pass was narrow, it resulted in a continuous pressure on the lower mountain sides. By September 2018, the mountain was deformed, and although the Yarlung Zangbo River did not break through, the river surface was severely blocked, eventually leading to the formation of a dammed lake (Figure 7).

## 4. Discussion

The avalanche and landslide and the river blocking dam disaster chain can be defined as: one or more geological hazards (such as a series of successive events in time, spatially dependent, interrelated, causal, and causing a chain reaction in sequence) that occur in succession [58,59,60]. This results in effectively a chain of disasters that block rivers due to avalanches, landslides, mudslides, and debris flow (Figure 8).

Landslides and the collapse of dams as a result of a series of earthquakes often occur in the form of a disaster chain. Generally, there are disaster chain models that are caused by single disasters and disaster chain models arising as a result of the coupling of multiple disasters. A kind of landslide and collapse disaster may be accompanied by other landslide and collapse disasters; or after the first kind of landslide disaster, another new kind of landslide disaster may be induced, that is, the first kind of landslide disaster reverted from the initial “damage ring” to the later “hazard inducing ring” [36,37,38].

The mechanistic path of the disaster chain may be represented as follows: glacial meltwater → mobilization of loose moraine → migration along the steep slope of the erosion trough to form glacier debris flow → formation of the dam plug to block the river channel → the dammed lake.

The long-term monitoring of the Sedongpu Basin and possible areas affected by formation of barrier lakes in the Yarlung Zangbo River should be strengthened. There are many loose deposits in the Sedongpu Basin, which have led to the blockage of the Yarlung Zangbo River many times throughout history. It is recommended that the Tibet Autonomous Region and relevant departments use high-resolution satellites, such as InSAR (Interferometric Synthetic Aperture Radar), for long-term monitoring of glaciers and gullies in the Sedongpu Basin [29,30]. Possible river blocking and risk assessment should be undertaken. It is also recommended that there should be construction of a high-standard automatic water level monitoring station in the vicinity of the upper reaches of the damming body for hydrological emergency monitoring and early warning work [40,41].

There should be further expansion of the geological and glacier surveys of the river section near the dammed body. In the ravine near the dammed body, there were also signs of glacial debris flow [44,45]. Based on a previous investigation, detailed geological disaster planning should be implemented, and specific relocation, monitoring or engineering treatment plans should be drawn up for potential major hazards.

An emergency response decision support platform for barrier lakes should be formulated such that there is a sharing of basic information on relevant land, hydrology, meteorology, earthquake, and other relevant factors [38]. Also a rapid risk assessment of the barrier lake including rapid analysis of the reservoir, upstream incoming water prediction, upstream flooding, damming object characteristics and stability, dam flood and downstream impact should be realized. In addition, a remote network consultation facility should be established to provide decision support for emergency response planning and implementation.

## 5. Conclusions

The warming rate of the Qinghai–Tibet Plateau is twice the rate for the global average. Against a background of rapid global warming, the instability of glaciers will increase, ablation will increase, and avalanche disasters will continue or even become more frequent; especially secondary disasters which will lead to the formation of disaster chains with the following features:

(1)There is a significant amplification effect in the avalanche chain.

The disaster chain will cause the disaster-affected body to suffer continuous damage in a short period of time. The lack of a holistic and systematic disaster prevention and mitigation system would not be able to cope with the situation of multi-disasters where the various functions in the disaster system are interrelated and form a strong net effect. The net effect will make the crisis spread rapidly, making the “amplification” of the effect of the disaster more apparent.

(2)The avalanche chain often shows a long chain effect.

A disaster chain with more than three "chain" sections is often referred to as a long chain. The most prominent feature of the long chain is that even if the front chain dies out, the rear chain is still likely to continue, and the disaster is difficult to stop. There are many reasons why a long chain of avalanches and landslides occur resulting in a river blocking dam disaster chain. The main reason is that there are many disaster sources, and natural disasters readily overlap with human disasters, resulting in amplification of the disaster.

(3)The avalanche chain can show a shrinkage effect.

If the disaster crisis is managed inappropriately, there will be an amplification of the disaster chain; if properly managed, however, there can be some control of the avalanche and landslide and the river blocking dam, so that the disaster chain will not continue, and thus the effect of the broken chain can be effectively reduced. Therefore, as a future research goal, it is imperative that we understand the mechanisms and processes of the avalanche, the landslide and the river blocking dam disaster chain, so that a comprehensive assessment of risk can be achieved in such a way that there is a rational allocation of disaster prevention and mitigation resources to ensure that the impact of avalanches and landslides is minimized.

## Figures and Tables

**Figure 1 ijerph-16-04707-f001:**
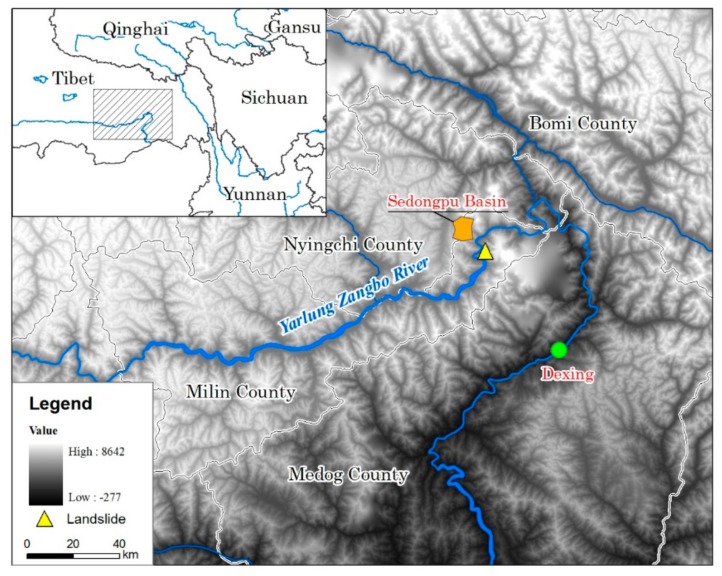
Location of the study area.

**Figure 2 ijerph-16-04707-f002:**
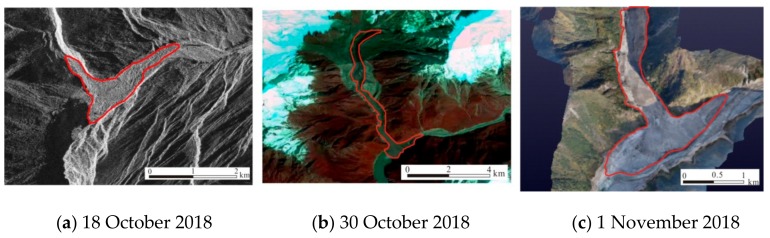
Comparison of remote sensing images before and after the collapse of the barrier lake.

**Figure 3 ijerph-16-04707-f003:**
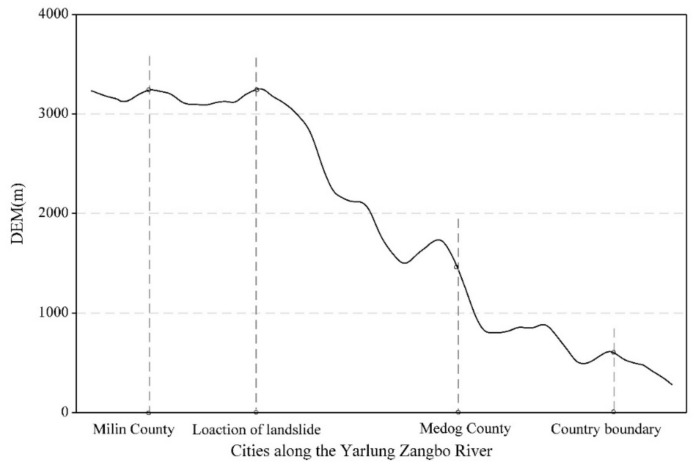
Digital elevation model (DEM) of the study area.

**Figure 4 ijerph-16-04707-f004:**
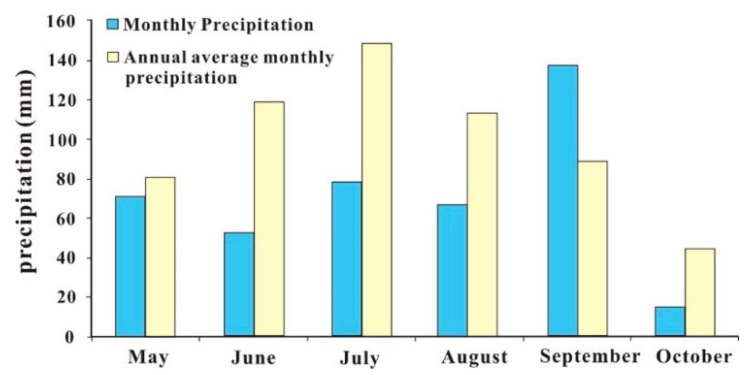
Monthly precipitation in Milin County from May to October 2018.

**Figure 5 ijerph-16-04707-f005:**
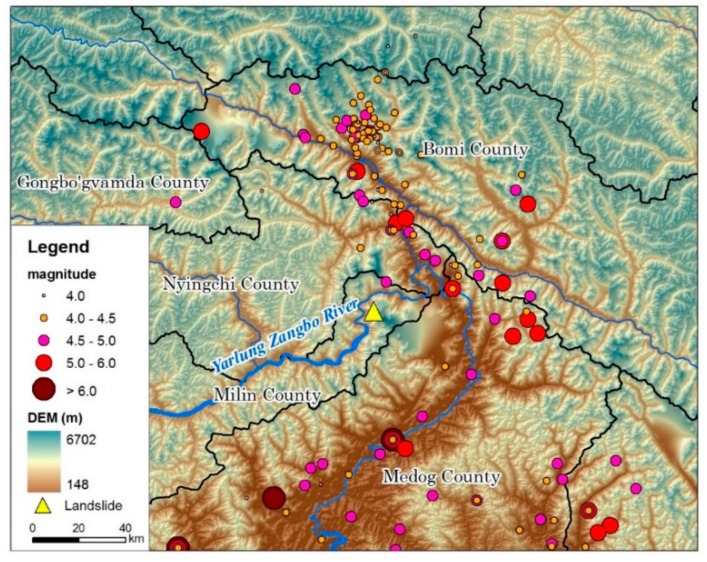
Recent earthquake distribution map for the study area (1949–2018).

**Figure 6 ijerph-16-04707-f006:**
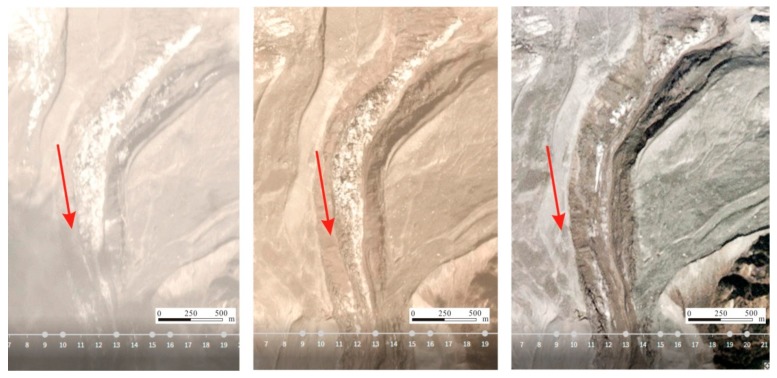
Comparison of remote sensing imagery on 26, 27, and 30 October 2017 for avalanche deposits in the Sedongpu Basin.

**Figure 7 ijerph-16-04707-f007:**
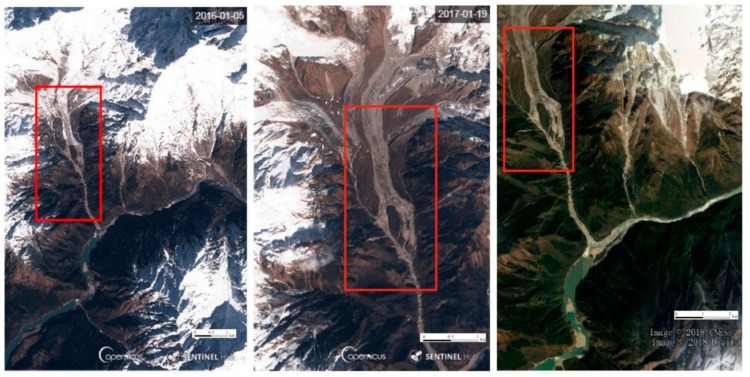
Changes in gully and sediment in the upper margin of the Yarlung Zangbo River (2016–2018).

**Figure 8 ijerph-16-04707-f008:**
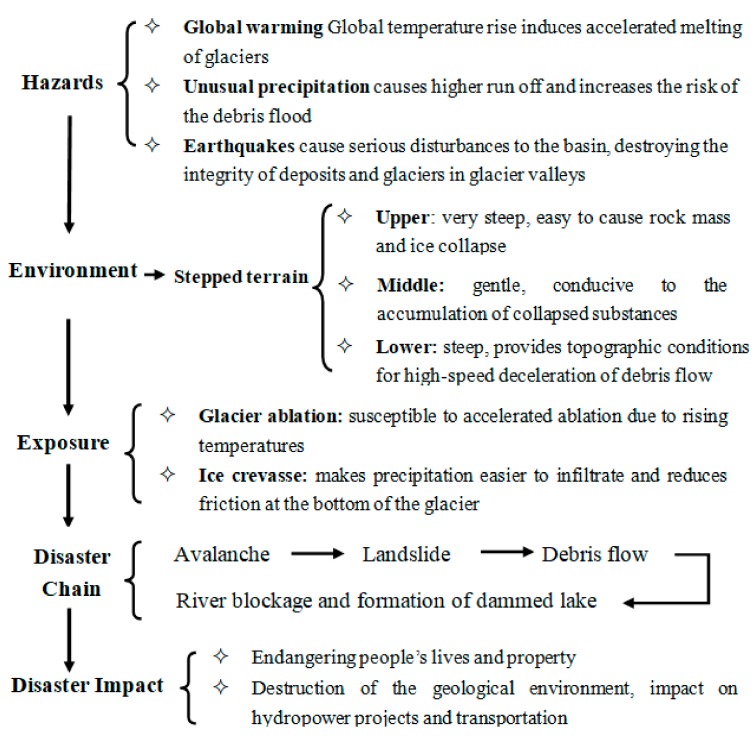
The evolution mechanism of the disaster chain (avalanche–landslide–debris flow–barrier lake).

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
