# Peer review of "Disaster Chain Analysis of Avalanche and Landslide and the River Blocking Dam of the Yarlung Zangbo River in Milin County of Tibet on 17 and 29 October 2018"

_ijerph, 2019, doi:10.3390/ijerph16234707_

Round 1
Reviewer 1 Report
The paper at hands deals with a disaster chain analysis including a landslide and a glacial avalanche within a couple of days, resulting in the damming of the Yarlung Zangbo river. Such polygenetic events in high mountain environments caused by a combined forcing of internal and external factors such as climate warming, earthquakes, steep topography and/or heavy rainfall become increasingly more relevant and thus more recognized by earth scientists due to climate change. So, generally this is an interesting topic, but I see a couple of points that I would like to see addressed by the authors prior to publication. Below are my recommendations to the authors, structured in general suggestions related to content and specific comments:
General remarks:
The chapter 1. Introduction does not provide the information an introduction is generally meant for. The authors give a lot of information about their study area, which is again presented in chapter 2, and thus redundant. I recommend to discuss the state of the art and the theoretical background on disaster chain analysis including relevant references from other high mountain environments in chapter 1. Introduction, and save the specific information about the Yarlung Zangbo study site for chapter 2. The authors highlight two different and distinct events within a couple of days, a landslide creating a huge natural dam blocking the Yarlung Zangbo river on October 17, and a glacial debris flow on October 29. Right now, from my point of view, the interrelation of these two events does not become sufficiently clear. In Chapter 2.1. Study area and data, average values on change of temperature, precipitation and permafrost features since the second half of the last century are presented. All over the world an amplification of climate change during the most recent decades can be observed and I am wondering whether similar trends are not present in the study area. If so, I recommend not to give average values over the whole period, for which data are available, but also shorter-term averages, for let’s say the last decade. In chapter 2.2 elevations are given with decimals (4,100.00) which should be omitted. In chapter 2.2 particular locations are mentioned (e.g. Dexing hydrological station) which should be indicated in Figure 1 and/or 2, 7, 8. The same holds true for the origin of the glacial debris flow. In general, Figures 2, 7 and 8 are very hard to read. Without scale bar distances/dimensions cannot be obtained, and particular features mentioned in the text should be highlighted by arrows, circles etc. Also, the figure captions should be improved, to give the reader the relevant information what can be depict by the images. A methods section is totally missing and must be added Titles of subchapters 3.1, 3.2, 3.3, and 3.4 are very long and should be replaced by shorter, more handy ones. In addition, some of these chapters are very short, so I recommend to generally restructure chapter 3. Chapter 4. Discussion actually is not a true discussion of the own results, rather recommendations are given what should be done in the future. I recommend to move chapter 3.5 to the beginning of the discussion chapter 4., as this chapter develops the disaster chain based on the previous results. Afterward the recommendations for future action can be made.
Specific remarks:
L42: longitudinal slope: what does this mean?
L53: environmental damage and economical disasters; instead of damage I recommend to use the term disturbance
L118: ratio: what does that mean
L130: 725.59 m: this value does not correspond to the values given above, please check.
L161-162: “…the recent precipitation in the southeastern part of the Qinghai-Tibet Plateau was significantly higher than that in the same period of the previous year, and in some areas it was four times greater”. I do not understand, according to Fig 4 and 5 its lower
L168: Figures 4 and 5 are very small and the quality is poor, lettering is barely readable
L173: Zuogong and Mangkang County not indicated in Figure 6
L178: topographical climate: what does this mean
L185: Figure 6 needs a legend for heights
L187: ice fissures: do you mean crevasses?
L189: integrated: I do not understand, what you mean.
L191: very rich: I also do not understand, what you mean.
L201-205: “The top of the Yarlung Zangbo River landslide dam had a large landslide from January 5, 2016 to June 8, 2018. The snow-covered sediment had been transported to the Yarlung Zangbo River, partially deposited in the middle of the mountain, because the lower mountain pass was narrow, and resulted in a continuous pressure on the mountains on the lower sides”: these two snetences are totally unclear to me.
L226: the term marine glaciers is definitely wrong. I guess you mean maritime, but even this one I recommend to avoid, better use temperate glaciers
Author Response
Reviewer 1
The paper at hands deals with a disaster chain analysis including a landslide and a glacial avalanche within a couple of days, resulting in the damming of the Yarlung Zangbo River. Such polygenetic events in high mountain environments caused by a combined forcing of internal and external factors such as climate warming, earthquakes, steep topography and/or heavy rainfall become increasingly more relevant and thus more recognized by earth scientists due to climate change. So, generally this is an interesting topic, but I see a couple of points that I would like to see addressed by the authors prior to publication. Below are my recommendations to the authors, structured in general suggestions related to content and specific comments:
Reply:
Thank you for your encouragement and concern. We are trying hard to improve our manuscript. We appreciate for your warm review work earnestly. The comments are all valuable and very helpful for revising and improving our paper, as well as the important guiding significance to our researches. We have studied comments carefully and have made correction which we hope meet with approval. Once again, thank you very much for your good comments and suggestions.
General remarks:
The chapter 1. Introduction does not provide the information an introduction is generally meant for. The authors give a lot of information about their study area, which is again presented in chapter 2, and thus redundant. I recommend to discuss the state of the art and the theoretical background on disaster chain analysis including relevant references from other high mountain environments in chapter 1. Introduction, and save the specific information about the Yarlung Zangbo study site for chapter 2. The authors highlight two different and distinct events within a couple of days, a landslide creating a huge natural dam blocking the Yarlung Zangbo river on October 17, and a glacial debris flow on October 29. Right now, from my point of view, the interrelation of these two events does not become sufficiently clear.
Reply:
We greatly appreciate the reviewers' comment. Thank you for pointing this out. We have added summary of the state of the art and the theoretical background on disaster chain analysis including relevant references from other high mountain environments in chapter 1 (Page 1, line 35-42; Page 2, line 43-93; Page 3, line 94-110). Also, we have saved the specific information about the Yarlung Zangbo study site for chapter 2 (Page 4, line 176-181). The interrelation of these two events was sufficient. Two major landslide disasters occurred in the same place twice in such a short time interval. And The Yarlung Zangbo River is an international river and the discharge of the barrier lake will affect downstream regions in India. Inappropriate management of the river may lead to international disputes. Moreover, disaster emergency control measures may be severe, population control, transfer and resettlement are usually challenging, and the rise and fall of the dam water would be likely to cause secondary disasters, which would seriously threaten the upstream and downstream residential areas and major infrastructure. So it is of great significance to take the two different events within a couple of days as a case study in the manuscript.
In Chapter 2.1. Study area and data, average values on change of temperature, precipitation and permafrost features since the second half of the last century are presented. All over the world an amplification of climate change during the most recent decades can be observed and I am wondering whether similar trends are not present in the study area. If so, I recommend not to give average values over the whole period, for which data are available, but also shorter-term averages, for let’s say the last decade.
Reply:
We agree with this comment. The existing available meteorological records on the Tibetan Plateau are generally shorter. The warming rate at the Tibetan Plateau was 0.3℃ 10y-1 over the past three decades, which was twice the rate of global warming. According to the meteorological record since 1961, the Qinghai-Tibet Plateau is gradually becoming "wet" in the context of global climate warming. The plateau has become "wet", with an increase in precipitation of 4.5 mm per 10 years in the rainy season and 1.8 mm per 10 years in the dry season (Page 4, line 151-159). We are also concerned about glacier changes. The overall area of the Purog Kangri Glacier has been reduced significantly over the period 1973 to 2016. In 2016, the area of the Purog Kangri Glacier was 389.0 km2, a decrease of 17.9% compared with 1973. We have made the corresponding adjustment in the revised manuscript (Page 4, line 148-166).
In chapter 2.2 elevations are given with decimals (4,100.00) which should be omitted. In chapter 2.2 particular locations are mentioned (e.g. Dexing hydrological station) which should be indicated in Figure 1 and/or 2, 7, 8. The same holds true for the origin of the glacial debris flow. In general, Figures 2, 7 and 8 are very hard to read. Without scale bar distances/dimensions cannot be obtained, and particular features mentioned in the text should be highlighted by arrows, circles etc. Also, the figure captions should be improved, to give the reader the relevant information what can be depict by the images.
Reply:
We agree with this comment. Thank you for pointing this out. We have omitted decimals of elevations (4,100.00) in the revised manuscript (Page 6, line 203-210). The location of Dexing hydrological station has been added in Figure 1 (Page 6, line 191-192). It is really true as reviewer pointed that Figures 2, 7 and 8 are very hard to read. We have highlighted the particular features mentioned in the text by circles, square box etc. in Figures 2 and 8. It is very difficult to mark in Figure 7. The figure captions have also been adjusted accordingly in the revised manuscript (Page 7, line 232-233; Page 11, line 300-302).
A methods section is totally missing and must be added Titles of subchapters 3.1, 3.2, 3.3, and 3.4 are very long and should be replaced by shorter, more handy ones. In addition, some of these chapters are very short, so I recommend to generally restructure chapter 3. Chapter 4.
Reply:
We greatly appreciate the reviewers' comment. Thank you for pointing this out. The method of this paper is based on typical cases, and analyzes disaster systems for disaster-causing mechanisms and disaster-derived chains. Through the induction and summary of the disaster process, a framework of chain mitigation and disaster reduction was constructed. Some of chapters are very short, so chapters 3.1 and 3.2 3.3 were merged into 3.1 (Page 8, line 245-257; Page 9-11). We have replaced shorter titles of subchapters 3.1, 3.2, and 3.3 in the revised manuscript (Page 7, line 236-237; Page 10, line 289).
Discussion actually is not a true discussion of the own results, rather recommendations are given what should be done in the future. I recommend to move chapter 3.5 to the beginning of the discussion chapter 4., as this chapter develops the disaster chain based on the previous results. Afterward the recommendations for future action can be made.
Reply:
Special thanks to you for your good comments. We agree with this comment. According to the Reviewer’s suggestion, we have moved chapter 3.5 to the beginning of the discussion chapter 4 in the revised manuscript (Page 12, line 322-337; Page 13, line 338-340).
Specific remarks:
L42: longitudinal slope: what does this mean?
Reply:
We greatly appreciate the reviewers' comment. Thank you for pointing this out. Vertical gradient should be more accurate. The slope of the longitudinal profile of a river (river valley, channel, etc.) is a basic morphometric measurement index reflecting the steps of each stage, which indirectly reflects the potential energy. The vertical gradient is a quantitative indicator reflecting the slope of longitudinal profile, namely: i=â–³Z∕L×1000‰ i: vertical gradient, often expressed as 1/1000; â–³Z: Height difference between the two points of the river bed and the bottom; L: horizontal distance between two points in the longitudinal direction. The term “Longitudinal slope” was replaced by “vertical gradient” but it was deleted in the revised manuscript (Page 3, line 104-105).
L53: environmental damage and economical disasters; instead of damage I recommend to use the term disturbance
Reply:
We agree with this comment. The term disturbance has been used in the revised manuscript (Page 3, line 117).
L118: ratio: what does that mean
Reply:
We greatly appreciate the reviewers' comment. Thank you for pointing this out. Ratio is not clear. Here should be gradient ratio. We have corrected it according to the Reviewer’s suggestion in the revised manuscript (Page 6, line 204).
L130: 725.59 m: this value does not correspond to the values given above, please check.
Reply:
We agree with this comment. We have checked it and unclear about this data source, so it was deleted in order to avoid misunderstanding in the revised manuscript (Page 7, line 217).
L161-162: “…the recent precipitation in the southeastern part of the Qinghai-Tibet Plateau was significantly higher than that in the same period of the previous year, and in some areas it was four times greater”. I do not understand, according to Fig 4 and 5 its lower
Reply:
We agree with this comment. Figure 4 does not describe the change in precipitation in Milin County, which is not suitable here. Therefore, we decided to delete Figure 4. It should be emphasized that in September, more precipitation occurred, in excess of 50% of normal levels, Figure 5 is correct. Corresponding changes were made in the revised manuscript (Page 8, line 255-257; Page 9, line 258-259).
L168: Figures 4 and 5 are very small and the quality is poor, lettering is barely readable
Reply:
We agree with this comment. We modified the lettering and improved the resolution for Figures 4 and 5 in the revised manuscript (Page 8, line 255-257).
L173: Zuogong and Mangkang County not indicated in Figure 6
Reply:
We greatly appreciate the reviewers' comment. Thank you for pointing this out. Figure 6 is a schematic view of a partial area of the Qinghai-Tibet Plateau. Zuogong and Mangkang County belong to Changdu City, which is far from these counties in the Fig.6 and are unable to be added. We have modified Figure 6 in the revised manuscript (Page 10, line 277-278).
L178: topographical climate: what does this mean
Reply:
We agree with this comment. This expression “topographical climate” is indeed inaccurate. We have made corrections in the revised manuscript (Page 9, line 268-269).
L185: Figure 6 needs a legend for heights
Reply:
We agree with this comment. A legend for heights has been added in Figure 6 in the revised manuscript (Page 10, line 277-278).
L187: ice fissures: do you mean crevasses?
Reply:
We agree with this comment. Thank you for pointing this out. We have changed ice fissures to crevasses in the revised manuscript (Page 10, line 279).
L189: integrated: I do not understand, what you mean.
Reply:
We greatly appreciate the reviewers' comment. Here “integrated” means “an integrated whole”. We have modified it in the revised manuscript (Page 10, line 281-282).
L191: very rich: I also do not understand, what you mean.
Reply:
We greatly appreciate the reviewers' comment. Here “very rich” means “there is a lot of …”. In order to avoid misunderstanding, this sentence was deleted in the revised manuscript (Page 10, line 283-284).
L201-205: “The top of the Yarlung Zangbo River landslide dam had a large landslide from January 5, 2016 to June 8, 2018. The snow-covered sediment had been transported to the Yarlung Zangbo River, partially deposited in the middle of the mountain, because the lower mountain pass was narrow, and resulted in a continuous pressure on the mountains on the lower sides”: these two snetences are totally unclear to me.
Reply:
We agree with this comment. Thank you for pointing this out. We have made some modifications of this sentence in the revised manuscript (Page 11, line 293-297).
L226: the term marine glaciers is definitely wrong. I guess you mean maritime, but even this one I recommend to avoid, better use temperate glaciers
Reply:
We agree with this comment. We greatly appreciate the reviewers' comment. Glacier ablation is more accurate in the revised manuscript (Page 13, line 338-340).
Reviewer 2 Report
Please see my specific comments and pay attention to comment 7:
1- Abstract: line 21 from "the formation conditions" to the end of the sentence is not clear and hard to follow, please break it down what this manuscript is proposing.
2- Line 40, is "capacity" means here the volume of the soil and dirt blocking the river? if so please revise it, it reads more like the volume of water being blocked behind the dam.
3- The third paragraph of introduction, between line 54 to 66, is a bit long and I suggest summarize this section and add more literature review of the studies that tried to implement disaster chain analysis, since Intro lacks context of methodology review.
4- A general comment, there are so many example like the following sentence that is hard to follow, because the sentence is too long. I would recommend an edit through the manuscript and try to break these sentences down into shorted sentences.
"Since 1961, the maximum depth of frozen soil in the Tibet region 4,500 m above sea level has been the most obvious sign, with an average decrease of 15.1 cm per 10 years."
5- There are many sentences that have been reworded and repeated through manuscript (e.g. first paragraph of Introduction and first paragraph of section 2.2). Since the manuscript is short, I would recommend deleting such redundancies. in general section 2.2 explains a lot of facts that have been discussed in introduction.
6- The titles and subtitles in section 3 are very long and can easily be shortened.
7- In general the manuscript is written in a report format rather than an academic journal paper. As you can see most of my comments are more about the formatting. I recommend a thorough revision on the manuscript and adding more discussion and scientific context to the manuscript. In it's current format, it is more like a geological report on an incident and does not have qualities to be considered as an article in an academic journal. My other recommendation is, just summarize the text into 2-3 pages and submit it as a short communication letter or editorial note.
Author Response
Reviewer 2
Please see my specific comments and pay attention to comment 7:
Abstract: line 21 from "the formation conditions" to the end of the sentence is not clear and hard to follow, please break it down what this manuscript is proposing.Reply:
Special thanks to you for your comments. It is really true as reviewer suggested that the sentence is not clear and hard to follow. We have broken it down and made some modifications according to the Reviewer’s comments (Page 1, line 21-25).
Line 40, is "capacity" means here the volume of the soil and dirt blocking the river? if so please revise it, it reads more like the volume of water being blocked behind the dam.
Reply:
We agree with this comment. Thank you for pointing this out. Here "capacity" means here the volume of the soil and dirt blocking the river. We have modified it in the revised manuscript (Page 6, line 198).
The third paragraph of introduction, between line 54 to 66, is a bit long and I suggest summarize this section and add more literature review of the studies that tried to implement disaster chain analysis, since Intro lacks context of methodology review.
Reply:
We greatly appreciate the reviewers' comment. Thank you for pointing this out. We have added summary of the state of the art and the theoretical background on disaster chain analysis including relevant references from other high mountain environments in the revised manuscript ((Page 1, line 35-42; Page 2, line 43-93; Page 3, line 94-110)).
A general comment, there are so many example like the following sentence that is hard to follow, because the sentence is too long. I would recommend an edit through the manuscript and try to break these sentences down into shorted sentences.
"Since 1961, the maximum depth of frozen soil in the Tibet region 4,500 m above sea level has been the most obvious sign, with an average decrease of 15.1 cm per 10 years."
Reply:
We agree with this comment. Thank you for pointing this out. We have modified it in the revised manuscript (Page 4, line 170-171). With the help of Charlesworth Author Services ([email protected]), a native English speaker has rechecked the language carefully throughout. We believe that the presentation of this revised manuscript has been greater improved, especially the English writing. Many syntactical errors in the whole text have been carefully corrected
There are many sentences that have been reworded and repeated through manuscript (e.g. first paragraph of Introduction and first paragraph of section 2.2). Since the manuscript is short, I would recommend deleting such redundancies. in general section 2.2 explains a lot of facts that have been discussed in introduction.
Reply:
We greatly appreciate the reviewers' comment. Thank you for pointing this out. The case in the introduction is only a brief introduction, which leads to the next study. Section 2.2 is the evolution process of the disaster chain case. We agree with this comment. First paragraph of Introduction and first paragraph of section 2.2 were simplified and some redundancies were deleted in the revised manuscript (Page 3, line 94-110; Page 4, line 149-151; Page 6, line 194-202).
The titles and subtitles in section 3 are very long and can easily be shortened.
Reply:
We agree with this comment. Thank you for pointing this out. We have shortened the titles and subtitles in section 3 in the revised manuscript (Page 7, line 234-237; Page 10, line 289).
7- In general the manuscript is written in a report format rather than an academic journal paper. As you can see most of my comments are more about the formatting. I recommend a thorough revision on the manuscript and adding more discussion and scientific context to the manuscript. In it's current format, it is more like a geological report on an incident and does not have qualities to be considered as an article in an academic journal. My other recommendation is, just summarize the text into 2-3 pages and submit it as a short communication letter or editorial note.
Reply:
Thank you for your encouragement and concern. We are trying hard to improve our manuscript. We appreciate for your warm review work earnestly. The comments are all valuable and very helpful for revising and improving our paper, as well as the important guiding significance to our researches.
The objective of this study is two-fold: (1) to analyze the “avalanche-landslide-debris flow-barrier lake” hazard mechanism and derivative disasters chain based on disaster system theory, and (2) to summarize the evolution process of the disaster chain and the organizing framework of chain-cutting disaster mitigation. These contributions may provide new insights into the disaster chain prevention and mitigation in the Qinghai-Tibet Plateau and other high mountain environments. (Page 3, line 135-140)
We have added summary of relevant references on the theoretical background and methods on disaster chain analysis from other high mountain environments in chapter 1 (Page 1, line 35-42; Page 2, line 43-93; Page 3, line 94-110). Also, we have saved the specific information about the Yarlung Zangbo study site for chapter 2 (Page 4, line 176-181). The method of this paper is based on typical cases, and analyzes disaster systems for disaster-causing mechanisms and disaster-derived chains. Through the induction and summary of the disaster process, a framework of chain mitigation and disaster reduction was constructed. Some of chapters are short, so chapters 3.1 and 3.2 3.3 were merged into 3.1 (Page 8, line 245-257; Page 9-11). Chapter 3.5 has been moved to the beginning of the discussion chapter 4 in the revised manuscript (Page 12, line 322-337; Page 13, line 338-340). So, afterward the recommendations for future action can be made.
We have studied comments carefully and have made correction which we hope meet with approval. Once again, thank you very much for your good comments and suggestions.
Reviewer 3 Report
Comments
Congratulation to the authors for developing this important manuscript. I believe this article will help the relevant stakeholders to have a better preparedness to deal with future disasters in the region. This however, there are few issues that need to be addressed.
Line 2 – the title is too long.
Line 32 – Introduction. I would be better if you could also briefly explain about the importance of disaster chain risk and how this approach has been useful in disaster preparedness planning (provide some evidence – literature)
Line 50-51 – potential English issues here.
Line 53 – be specific, what kind of environmental damage that might happen?
Line 77 – study area, would be useful if this section also discussed the disaster profile (history of relevant disaster events) of the research setting.
Line 229 – Discussion. It seems that the discussion is still too shallow. You need to be more in-depth in discussing the findings. The discussion should also confirm and maybe contrast your findings with the existing literature. Maybe you could also compare to what happened in the other part of the world that has similar context with your research. This is important to enrich your discussion.
Literature – it looks like that almost all your literature is from China. Maybe it would be great if you could also use important/relevant grey literature such as from the IPCC SREX, (https://www.ipcc.ch/site/assets/uploads/2018/03/SREX_Full_Report-1.pdf); or this one maybe (https://reliefweb.int/sites/reliefweb.int/files/resources/managing_disaster_risks_and_water_under_climate_change_in_ca_and_caucasus-compilation.pdf),; and also article from different countries such as this one “Landslide in changing climate” (https://www.sciencedirect.com/science/article/pii/S0012825216302458).
Author Response
Reviewer 3
Comments
Congratulation to the authors for developing this important manuscript. I believe this article will help the relevant stakeholders to have a better preparedness to deal with future disasters in the region. This however, there are few issues that need to be addressed.
Reply:
Thank you for your good comments and suggestions. We appreciate for your warm review work earnestly. The comments are all valuable and very helpful for revising and improving our paper, as well as the important guiding significance to our researches. We have studied comments carefully and have made correction which we hope meet with approval.
Line 2 – the title is too long.
Reply:
We agree with this comment. Thank you for pointing this out. The current manuscript title covers the most important research elements and issues and it is difficult to be short. We have shortened the titles and subtitles such as section 3 in the revised manuscript (Page 7, line 234-237; Page 10, line 289).
Line 32 – Introduction. I would be better if you could also briefly explain about the importance of disaster chain risk and how this approach has been useful in disaster preparedness planning (provide some evidence – literature)
Reply:
We greatly appreciate the reviewers' comment. A briefly explain about the importance of disaster chain risk and how this approach has been useful in disaster preparedness planning have been added in the revised manuscript (Page 1, line 35-42; Page 2, line 43-93; Page 3, line 94-110).
Line 50-51 – potential English issues here.
Reply:
We agree with this comment. We have modified it in the revised manuscript (Page 3, line 112-113).
Line 53 – be specific, what kind of environmental damage that might happen?
Reply:
We agree with this comment. Environmental damage may not be accurate. The term disturbance has been used in the revised manuscript (Page 3, line 117).
Line 77 – study area, would be useful if this section also discussed the disaster profile (history of relevant disaster events) of the research setting.
Reply:
We greatly appreciate the reviewers' comment. Thank you for pointing this out. The study area is characterized by its complex geology, high in-situ stresses, and earthquakes, repeating ground freeze and thaw, intensive precipitation etc., frequent landslides and chained disasters. The disaster profile (history of relevant disaster events) has been added in the revised manuscript (Page 4, line 148-189).
Line 229 – Discussion. It seems that the discussion is still too shallow. You need to be more in-depth in discussing the findings. The discussion should also confirm and maybe contrast your findings with the existing literature. Maybe you could also compare to what happened in the other part of the world that has similar context with your research. This is important to enrich your discussion.
Reply:
We agree with this comment. Chapter 3.5 has been moved to the beginning of the discussion chapter 4 in the revised manuscript (Page 12, line 322-337; Page 13, line 338-340). So, afterward the recommendations for future action can be made.
Literature – it looks like that almost all your literature is from China. Maybe it would be great if you could also use important/relevant grey literature such as from the IPCC SREX, (https://www.ipcc.ch/site/assets/uploads/2018/03/SREX_Full_Report-1.pdf); or this one maybe (https://reliefweb.int/sites/reliefweb.int/files/resources/managing_disaster_risks_and_water_under_climate_change_in_ca_and_caucasus-compilation.pdf),; and also article from different countries such as this one “Landslide in changing climate” (https://www.sciencedirect.com/science/article/pii/S0012825216302458).
Reply:
We greatly appreciate the reviewers' comment. Special thanks to reviewers for providing such an important reference. The references have been added in the revised manuscript (Page 1, line 35-40; Page 14, line 405-407; Page 15, line 408-412).
Round 2
Reviewer 1 Report
Besides some specific remarks, in the first round of review I had a couple of major concern, including the following: chapter 1. “Introduction” does not provide the information an introduction is generally meant for; missing information in chapter 2.1 “Study area and data”; recommendations to restructure chapter 3 and chapter 4.
Most of my specific remarks were followed, the list of references is much more comprehensive now and also my major concerns mentioned above were reasonably addressed, so from my point of view the manuscript now warrants publication.
I have only some more minor comments, which, from my point of view, would help to make the paper more enjoyable:
L 47: delete: “In the journal Nature,” and start sentence with: From the perspective of …
L 129: instead of “wet” better use slightly more humid
As already mentioned in my earlier review to me Figures 2, 6 and 7 are very hard to read, without scale bar, as distances/dimensions cannot be obtained, and particular features mentioned in the text should be highlighted by arrows, circles etc. None of these earlier suggestions by myself were followed in the revised version but I still believe them as very useful for the readers, so I ask the authors to consider to include the respective changes.
Author Response
Besides some specific remarks, in the first round of review I had a couple of major concern, including the following: chapter 1. “Introduction” does not provide the information an introduction is generally meant for; missing information in chapter 2.1 “Study area and data”; recommendations to restructure chapter 3 and chapter 4. Most of my specific remarks were followed, the list of references is much more comprehensive now and also my major concerns mentioned above were reasonably addressed, so from my point of view the manuscript now warrants publication.
Reply:
Special thanks for your good comments and suggestions. We appreciate for your warm review work earnestly again. The comments are all valuable and very helpful for revising and improving our paper, as well as the important guiding significance to our researches. We have studied comments carefully and have made correction which we hope meet with approval.
I have only some more minor comments, which, from my point of view, would help to make the paper more enjoyable:
L 47: delete: “In the journal Nature,” and start sentence with: From the perspective of …
Reply:
We agree with this comment. Thank you for pointing this out. We have deleted: “In the journal Nature,” and start sentence with: From the perspective of … in the revised manuscript (Page 2, line 47-48).
L 129: instead of “wet” better use slightly more humid
Reply:
We agree with this comment. Thank you for pointing this out. We have changed “wet” into “humid” in the revised manuscript (Page 3, line 132-134).
As already mentioned in my earlier review to me Figures 2, 6 and 7 are very hard to read, without scale bar, as distances/dimensions cannot be obtained, and particular features mentioned in the text should be highlighted by arrows, circles etc. None of these earlier suggestions by myself were followed in the revised version but I still believe them as very useful for the readers, so I ask the authors to consider to include the respective changes.
Reply:
We greatly appreciate the reviewers' comment. Thank you for your reminding again. It is really true as reviewer pointed that Figures 2, 6 and 7 are very hard to read. All figures (Fig. 2, 6 and 7) have been added scales and corresponding arrows, squares etc. in the revised manuscript (Page 5, line 202-204; Page 8, line 249-252; Page 9, line 264-267).
Reviewer 2 Report
I would like to thank authors for revising the manuscript thoroughly. In its current format, I can recommend the manuscript for publication at IJERPH. However, I have one major comment.
In figure 4, authors pointed out that precipitation was much higher than annual average (almost twice), which logically causes higher run off and increases the risk of the debris flood, but it is not included in the figure 8 as a hazard and just earthquake and global warming are mentioned as hazards. I was wondering if there is a reason that unusual precipitation is not considered as a hazard or there was any reasoning behind that. If not, I believe that should be added to the hazard list as well.
Author Response
I would like to thank authors for revising the manuscript thoroughly. In its current format, I can recommend the manuscript for publication at IJERPH. However, I have one major comment.
In figure 4, authors pointed out that precipitation was much higher than annual average (almost twice), which logically causes higher run off and increases the risk of the debris flood, but it is not included in the figure 8 as a hazard and just earthquake and global warming are mentioned as hazards. I was wondering if there is a reason that unusual precipitation is not considered as a hazard or there was any reasoning behind that. If not, I believe that should be added to the hazard list as well.
Reply:
We greatly appreciate the reviewers' comment. Thank you for pointing this out. Higher precipitation will really cause higher run off and increases the risk of the debris flood. It is very certain that we recognize this fact mentioned in the text. It should be included in the figure 8 as a hazard. We have added it to the hazard list as well in the revised manuscript ((Page 10, line 284-287).
Special thanks for your good comments and suggestions. We appreciate for your warm review work earnestly again. The comments are all valuable and very helpful for revising and improving our paper, as well as the important guiding significance to our researches.
Round 3
Reviewer 2 Report
I would like to recommend the manuscript to be considered for publication at IJERPH.